# Improving quality of empirical Greens functions, obtained by cross-correlation of high-frequency ambient seismic noise

Nikita Afonin[1,4], Elena Kozlovskaya[1,2], Jouni Nevalainen[1,3], Janne Narkilahti[3]

[1]Oulu Mining School, POB-3000, FIN-90014, University of Oulu, Finland
[2]Geological Survey of Finland, P.O. Box 96, FI-02151, Espoo, Finland
[3]Sodankylä Geophysical Observatory POB 3000, FIN-90014, University of Oulu, Finland
[4]N. Laverov Federal Centre for the Integrated Arctic Research RAS, Arkhangelsk, Russia

*Correspondence to*: Nikita Afonin (nikita.afonin@oulu.fi)

**Abstract.** Studying the uppermost structure of the subsurface is a necessary part for solving many practical problems
(exploration of minerals, groundwater studies, geoengineering, etc.). Practical application of active seismic methods for these purposes is not always possible because of different reasons, such as logistical difficulties, high cost of work, high level of seismic and acoustic noise. That is why developing and improving passive seismic methods is one of the important problems in applied geophysics. In our study, we describe the way of improving quality of Empirical Green's Functions (EGFs), evaluated from high-frequency ambient seismic noise, by using advanced technique of cross-correlation functions stacking in
the time domain (in this paper we use term "high-frequency" for the frequencies higher than 1 Hz). The technique is based on global optimization algorithm, in which the optimized objective function is a signal-to-noise ratio of an EGF, retrieved at each iteration. In comparison to existing techniques, based, for example, on weight-stacking of cross-correlation functions, our technique makes it possible to increase significantly the signal-to-noise ratio and, therefore, the quality of the EGFs. The technique has been tested with the field data acquired in an area with high level of industrial noise (Pyhäsalmi Mine, Finland)
and in an area with low level of anthropogenic noise (Kuusamo Greenstone Belt, Finland). The results show that the proposed technique can be used for extraction of EGFs from high-frequency seismic noise in practical problems of mapping of the shallow subsurface, both in areas with high and low level of high-frequency seismic noise.

## 1 Introduction

Seismic methods as tools for studying the shallow subsurface structures in exploration geophysics have been developed during
many years. Traditionally, seismic surveys (reflection and refraction) have been carried out using active sources. The reflection and refraction controlled-source seismic sounding methods are widely applied in exploration for oil and gas, but less commonly in mineral exploration in crystalline bedrock areas. The reasons for this have been the traditionally high cost of seismic surveys and logistical difficulties (Malehmir et al., 2012). Seismic methods as a mineral exploration tool are very good for delineation of the boundaries of certain types of mineral deposits as well as for estimating their ore potential (Kukkonen et al., 2009;
Malehmir et al., 2012). There are, however, challenges in exploration of new deep targets in the vicinity of active mines, that is, in brownfield exploration. In our paper, brownfield means exploration near active mines or at the previously studied area with the purpose of getting new mineral reserves, while greenfield means exploration of new mineral deposits. Due to the large amount of heavy machinery, the active mines themselves produce strong seismic and acoustic noise. This continuous noise is overlapping in frequencies with the signals of the controlled seismic sources, creating a problem for the high-resolution active-
source seismic experiments in a brownfield exploration (Place et al., 2015).

In our paper, we describe results of investigating the possibility to use passive ambient seismic noise interferometry with the noise with frequencies higher than 1 Hz (hereafter we use the term "high-frequency" for this seismic noise) for extracting information about the shallow subsurface in greenfield and brownfield exploration projects. In our study, the shallow subsurface means depths from ground surface down to several hundreds meters. For this, we develop a new method of

improving quality of empirical Green's function (EGFs) evaluated from high-frequency industrial, anthropogenic or natural seismic noise. We partly use algorithms described in Campillo (2006), Bensen et al. (2007), Groos et al., (2012), Poli et al. (2012a, 2012b, 2013), Afonin et al. (2017) for ambient noise pre-processing and implement a new algorithm of stacking cross-correlation functions in the time domain.

At present, there are several advanced algorithms, working in time domain, in frequency domain or in time-frequency domain.
The one group of algorithms tries to improve the quality of the resulting EGFs using evaluation of cross-correlation functions according to some criteria prior to stacking them. For example, in the methods described in Baig et al. (2009), a denoising procedure, based on S-transform, is applied to cross-correlation functions before their stacking. In the "time-frequency domain phase-weighted stacking" method, which may use either S-transform (Shimmel et al., 2011), or wavelet transform (Ventosa et al., 2017), phases of signal are analysed prior stacking them. The errors that inverse S-transforms may introduce to subsequent
phase-velocity measurements were analysed in Li et al. (2018). Another approach, based on stacking only cross-correlation functions of highly coherent signals was used in global scale coda wave interferometry studies (Boué et al., 2014). These algorithms are not using signal-to-noise ratio (SNR) of cross-correlation functions for improving the final EGF, and it is assumed that signal coherence by itself is a guarantee that all non-suitable cross-correlation functions are either excluded from the final stack or there is minimised by using weights and hence the SNR is automatically improving. This may be true for
teleseismic coda wave interferometry (Pham et. al., 2018), in which source location is a-priori known and it is easy to control that only signals within so-called "stationary phase" area are cross-correlated (Wapenaar et al., 2010). However, in the ambient noise studies with stochastically, non-evenly distributed in time and space noise sources, the azimuthal distribution of them is not known a-priory. In this case, one would need to take into account this distribution, in order to satisfy the stationary phase condition. The other group of methods, such as "root mean square stacking" or weight stacking (Shirzad et al., 2014, Nakata
et. al., 2015, Cheng et al., 2015, Li et al. 2018) are aiming mainly to increase a signal-to-noise ratio of the resulting EGF, but they do not take into account coherence of the cross-correlation functions in the stack. That is why incoherent cross-correlation functions are not totally excluded from stack in these algorithms and this can decrease the quality of evaluated EGFs.

To overcome limitations of existing techniques, we develop a new algorithm that makes it possible not only to exclude incoherent cross-correlation functions from EGFs stacking process, but also to keep control on azimuthal distribution of noise
sources and condition of "stationary phase". In our paper the term 'coherent' is used to define cross-correlation functions with the same time lags of signal maxima and the same dominant frequency. We do not use this method in the frequency domain because for stationary phase condition to be satisfied it is important to stack cross-correlation functions with the same time lags and dominant frequencies, in other words, functions that are coherent to each other. As a main criterium for selecting cross-correlation functions to stack, we use increase of SNR of extracted EGFs after stacking. Moreover, we use global

optimization algorithm for obtaining the best solution for SNR. In this case, a SNR, calculated on each iteration, is an objective function that is optimized.

In our paper, we are presenting details of this algorithm and illustrate its performance using passive seismic ambient noise data acquired in two areas of Fennoscandia: Pyhäsalmi mine (as an example of area with high level of industrial noise) and Kuusamo
Greenstone Belt area (quiet area prospective for new mining projects (Wiehed et al., 2005; Lehtonen et al., 2009).

## 2 Advanced technique of cross-correlation functions stacking

For solving the problems described in the Introduction, we suggest our method of time-domain stacking of cross-correlation functions calculated for different time windows. We call this method signal-to-noise ratio (SNR) stacking. The general purpose of this method is to select for stacking only those cross-correlation functions that are not only coherent to each other, but also
correspond to the stationary phase area.

Let us assume that ambient noise in some frequency band is recorded simultaneously at two different points with Cartesian coordinates $r_1$ and $r_2$ . For each frequency, the stationary phase area for the receiver located in the point $r_i$, $i=1,2$ corresponds to Fresnel zone of the wave propagating from the source to the receiver with some apparent velocity. In this case, the maximum of the cross-correlation function at some time lag would correspond to the minimum of apparent velocity and hence, the cross-
correlation function would be close to the "true" EGF. We assume that noise sources are partly located in a stationary phase area while other noise sources are distributed outside it. For selection of cross-correlation functions corresponding to the stationary phase area, it is possible to use criteria of minimum apparent velocity and of the signal-to-noise ratio increasing after stacking. We consider the SNR of EGF after stacking as some generally non-linear function of apparent velocity and backazimuth of noise sources and an initial time window used to start selection of cross-correlation functions to the stack. In
this case, the global optimisation of this objective function would allow to retrieve EGFs of high quality.

We assume again that the ambient seismic noise is recorded simultaneously at two different points with Cartesian coordinates $r_1$ and $r_2$, $r = [x, y, z]$ and continuous recordings are split into $n$ time windows with the same durations. Let $a_i(r_1, r_2, t)$ is the cross-correlation function of these seismic records for the time window $i$, $i = 1 \dots n$, where $t$ is a time lag of the seismic records. Let $t_m$ is the maximum time lag in a cross-correlation function (length of cross-correlation); $t_{ds}$ is a maximum time
of wave propagation between the two points; $|t_m| \gg |t_{ds}|$ and $-t_m \leq t \leq t_m$. Let $-t_{ds} \leq t_e \leq t_{ds}$ is the time lag on the cross-correlation function corresponding to the expected seismic phase (body or surface wave) and $\triangle t_e = t_e \pm T$, where $T$ is the period of expected signal. Negative values of the time lags correspond to the anti-casual part of the evaluated EGF. In this case, selection of $t_{ds}$ and $\triangle t_e$ is based upon *a priori* information about seismic velocities in the studied area. The value of $\triangle t_e$ is at least two periods of the expected signal dominant frequency. In the case of evaluation of surface wave parts of EGFs,

this frequency usually corresponds to the frequency of noise with the largest amplitude that can be estimated by time-frequency analysis of the seismic noise records.

Let $a_i^{max}(\boldsymbol{r_1}, \boldsymbol{r_2}, \Delta t_e)$ is the maximum value of cross-correlation function in the time interval $\Delta t_e$. Then, the signal-to-noise ratio of the cross-correlation function calculated for the $i\,th - $ time window $(SNR(a_i(\boldsymbol{r_1}, \boldsymbol{r_2}, t)))$ is:

$$SNR(a_i(\boldsymbol{r_1}, \boldsymbol{r_2}, t)) = \frac{a_i^{max}(\boldsymbol{r_1}, \boldsymbol{r_2}, \Delta t_e)}{\frac{1}{2|t_m - t_{ds}|}(\int_{t_{ds}}^{t_m} a_i^2(\boldsymbol{r_1}, \boldsymbol{r_2}, t)dt + \int_{-t_m}^{-t_{ds}} a_i^2(\boldsymbol{r_1}, \boldsymbol{r_2}, t)dt)} \qquad (1)$$

Let $a_i(\boldsymbol{r_1}, \boldsymbol{r_2}, t)$ and $a_j(\boldsymbol{r_1}, \boldsymbol{r_2}, t)$ are cross-correlation functions calculated for two different time windows $i \in (1..n)$ and $j \in (1..n)$ and $c(\boldsymbol{r_1}, \boldsymbol{r_2}, t) = a_i(\boldsymbol{r_1}, \boldsymbol{r_2}, t) + a_j(\boldsymbol{r_1}, \boldsymbol{r_2}, t)$ is an EGF retrieved from these two cross-correlation functions. If $a_i(\boldsymbol{r_1}, \boldsymbol{r_2}, t)$ and $a_j(\boldsymbol{r_1}, \boldsymbol{r_2}, t)$ are coherent to each other and $i \neq j$, then expressions $SNR(a_i(\boldsymbol{r_1}, \boldsymbol{r_2}, t)) < SNR(c(\boldsymbol{r_1}, \boldsymbol{r_2}, t))$ and $SNR(a_j(\boldsymbol{r_1}, \boldsymbol{r_2}, t)) < SNR(c(\boldsymbol{r_1}, \boldsymbol{r_2}, t))$ have to be true, according to the principle of interference. Condition $i \neq j$ is

10 necessary in order to avoid stacking of functions with itself. Therefore, increasing SNR of the retrieved EGF after stacking with some cross-correlation function can be used as a criterion for selection of this function to the stack, excluding incoherent functions from the stack and building up the EGF with high signal-to-noise ratio.

Based on the criteria described above, an expression for calculation of EGF for $k$-th iteration can be written as

$$G^k(\boldsymbol{r_1}, \boldsymbol{r_2}, t) = \sum_{\substack{i=1 \\ i \neq k}}^{n}(G_i^k(\boldsymbol{r_1}, \boldsymbol{r_2}, t) + a_i(\boldsymbol{r_1}, \boldsymbol{r_2}, t) * \delta(G_i^k, a_i)), \qquad (2)$$

15 where $k = 1 \dots n$ is the number of initial function; $n$ is the number of time windows; $i = 1, \dots, n$; $G_i^k(\boldsymbol{r_1}, \boldsymbol{r_2}, t)$ is EGF corresponding to $k$-th – initial function and evaluated in previous iterations:

$$G_i^k(\boldsymbol{r_1}, \boldsymbol{r_2}, t) = \begin{cases} a_k(\boldsymbol{r_1}, \boldsymbol{r_2}, t), & i = 1 \\ G_{i-1}^k(\boldsymbol{r_1}, \boldsymbol{r_2}, t), & i \neq 1 \end{cases}. \qquad (3)$$

The operator of selection can be written as

$$\delta(G_i^k, a_i) = \begin{cases} 0, SNR\left(G_i^k(\boldsymbol{r_1}, \boldsymbol{r_2}, t) + a_i(\boldsymbol{r_1}, \boldsymbol{r_2}, t)\right) < SNR\left(G_i^k(\boldsymbol{r_1}, \boldsymbol{r_2}, t)\right); \\ 1, SNR\left(G_i^k(\boldsymbol{r_1}, \boldsymbol{r_2}, t) + a_i(\boldsymbol{r_1}, \boldsymbol{r_2}, t)\right) \geq SNR\left(G_i^k(\boldsymbol{r_1}, \boldsymbol{r_2}, t)\right); \end{cases} \qquad (4)$$

20 As a result of this algorithm we obtain $n$ candidates for EGF that can be considered as solutions to the optimization problem in some parameter space. Let us consider the signal-to-noise ratio as some function $f(k)$, where $k$ is the index of initial functions: $SNR(G^k(\boldsymbol{r_1}, \boldsymbol{r_2}, t)) = f(k), k = 1, \dots, n$. Then the condition for the final EGF selection can be written as $m = argmax(f(k))$, where $m$ denotes the index of EGF selected to the stack. Following this condition, the EGF with maximum

signal-to-noise ratio will be selected as the final one. As the function $f(k)$ may have several local maxima in the parameter space $k$, $k=1,...,n$, the condition for the final EGF selection ensures that the global maximum of this function is obtained in the parameter space considered.

In the proposed algorithm, maximizing the signal-to-noise ratio of the retrieved EGF is ensured by stacking of only cross-correlation functions coherent to each other and selection of EGF with the maximum signal-to-noise ratio from all calculated candidate EGFs. In other words, the proposed algorithm is analogous to the direct search methods of global optimization. It is necessary to remember, however, that EGF with the maximum signal-to-noise ratio does not correspond to a true EGF, if the dominant noise sources are located outside the stationary phase area. Therefore, it is important to use the system of observations that allows estimating azimuthal distribution of noise sources. Moreover, the method is based on assumption that sources of the ambient seismic noise produce a signal with relatively broad bandwidth and cannot produce an ideal harmonic signal of single frequency.

The method also makes it possible to keep control over a-priory unknown azimuthal distribution of noise sources. For this, a 2-D array of seismic recording stations is necessary. In this case, the time lags, corresponding to expected signal $\Delta t_e$ in Eq. 1 have to be a function of apparent seismic velocity and backazimuth: $\Delta t_e = f(v, \varphi)$. Then signal-to-noise ratio for each pair of stations of the array is the function of initial function index, velocity and backazimuth: $SNR(G^k(\mathbf{r_1}, \mathbf{r_2}, t)) = f(k, v, \varphi), k = 1, ..., n, \ v_{min} \le v \le v_{max}, \ 0 \le \varphi \le 360$. Limits of apparent velocities have to be calculated according to a-priory information about seismic velocities in the studied area. A global maximum of the function corresponds to the strongest or the most coherent wavefield. Therefore, the method allows estimating azimuth to the strongest source of noise wavefield.

We suggest that this method can be used for extraction of EGFs from high-frequency industrial, anthropogenic, or natural seismic noise. Moreover, this method does not require that only a diffuse field is used for calculating EGFs. Therefore, application of this method to the data of optimally selected seismic recording array might decrease significantly the time necessary for registration of ambient seismic noise, which is very important for practical applications of passive seismic interferometry. For studying the possibilities of using this method for extraction of EGFs from high-frequency seismic noise, we use the data from two passive seismic experiments carried out in areas with different seismic noise characteristics. The first area is characterized by high level of industrial noise (Pyhäsalmi underground mine site) that is usually observed in brownfield exploration areas, while the second area is seismically very quiet and is characterized by a limited amount of local anthropogenic (roads) and natural (rivers) high-frequency seismic noise sources. Such noise characteristics are typical for greenfield exploration areas.

## 3 Experimental data

### 3.1 Pyhäsalmi mine area

As an example of using high level industrial seismic noise for estimation of EGFs, we used the seismic noise at the site of Pyhäsalmi mine, Finland. For this purpose we installed 24 3-component DSU-SA MEMS (microelectromechanical system)

seismic sensors with the autonomous RAUD eX data acquisition units manufactured by Sercel Ltd (France). The instruments were installed along a 10-km-long line crossing the mine area with interstation distances of about 100 m (for PLB03-PLB13 and PLB14-PLB22) and 2 km (PLB01, PLB02, PLB23, PLB24) (Figure 1). The seismic stations recorded continuous seismic data from 1.11.2013 to 5.11.2013 with a sampling frequency of 500 samples per second (sps).

The profile configuration was selected on the base of the test measurements of ambient noise in Pyhäsalmi area made by authors. These studies showed that the mine is the main source of seismic high-frequency noise at distances about several kilometres from the mine.

The profile crossing the mine area consists of two parts, and each of these consists of 12 sensors: the western part has direction from the mine to the West (PLB01-PLB13), and the eastern part has direction from the mine to the East (PLB14-PLB24). Each

part of the profile includes one sensor closest to the mine (PLB13 and PLB14). The horizontal components were oriented to the true North and East (NS and EW-components, respectively). Thus, rotation of the horizontal components before seismic noise analysis was not necessary.

### 3.2 Kuusamo Greenstone Belt area

As an example of an area with low level of anthropogenic seismic noise, we selected an area located in the Kuusamo

Greenstone Belt (KuGB), Finland, because of numerous previous geological and geophysical studies there (Silvennoinen, 1991; Bruneton et al., 2004; Yliniemi et al., 2004; Silvennoinen et al., 2007; Poli et al., 2012; Pedersen et al., 2013; Silvennoinen et al., 2014; Tiira et al., 2014; Vinnik et al., 2014; etc.). Moreover, Weighed et al. (2005) and Lehtonen et al. (2009) have shown that this area is prospective for gold- and diamond deposits.

For testing our method of cross-correlation function stacking, we use the data collected during a passive seismic experiment

in KuGB area in August and September 2014. One of the targets of this experiment was to investigate the possibility of high-frequency EGFs extraction from anthropogenic or natural seismic noise in regions with low ambient noise level.

The temporary seismic array (Figure 2) consisted of five three-component velocimeters Trillium Compact produced by Nanometrics (Canada) and 24 three-component accelerometers DSU-SA MEMS with autonomous RAUD eX data acquisition units manufactured by Sercel Ltd.

As one can see in Figure 2, the seismic array represents a triangle. The sides of this triangle are about 4-6 km long. The broadband (BB) sensors were installed in the vertices of this triangle and collocated with MEMS accelerometers. In addition, each of these large triangle vertices was surrounded by a circular array with small aperture (about 1400-1500 m), consisting of six accelerometers. The whole array recorded continuous seismic data from 28.08.2014 to 10.09.2014 with a sampling rate of 500 sps. Such an array configuration makes it possible to estimate the azimuthal distribution of the high-frequency noise

sources and also to extract high-frequency EGFs from records of small aperture arrays.

### 4 Analysis of the seismic noise

### 4.1 Time-frequency analysis

One of the most important steps of the data preparation before extraction of EGFs is the time-frequency analysis. It is necessary for selection of a frequency band with high amplitudes of the ambient noise. For this, we analysed characteristics of the seismic noise recorded at different distances from the potential noise sources. In the Pyhäsalmi experiment, the most probable noise sources are located inside the underground mine and in the open pit. For the time-frequency analysis of the seismic noise, we used records of sensors installed at different distances from the mine and from the open pit (PLB24 and PLB14 (Figure 1)). Figure 3 (a, b) shows the results of this analysis.

From Figure 3 (a, b), one can see two main frequency bands with high amplitudes of the seismic noise recorded closest to the mine: about 3-4 Hz and about 10-100 Hz, respectively. Moreover, the amplitudes of the noise in these frequency bands decrease with distance from the mine. Therefore, we can assume that the sources of the noise for these frequency bands are located inside the underground mine and in the open pit. Based on this analysis, we selected the frequency band of 1-100 Hz for pre-filtering of the noise prior to calculation of cross-correlation functions.

In the KuGB experiment, a temporary seismic network was installed in a quiet area without any significant industrial activity; therefore, we can assume that the high-frequency seismic noise might be produced by multiple natural (for example, rivers) and/or anthropogenic (for example, roads) sources. In this case, analysis of time-frequency characteristics of the seismic noise is a necessary step. For this, we calculated time-frequency diagrams in the frequency band of 0.1-100 Hz and examples of these diagrams for two stations are presented in Figure 3 (c, d)).

Figure 3 (c, d) shows that noise records of both stations have amplitude maximums in the frequency band of 0.1-1 Hz. Seismic noise recorded by KU05 station is also characterized by periodically high amplitudes in the frequency band of 40-100 Hz (Figure 3 (c)). This noise may be caused by anthropogenic (transport) or natural (for example, wind) sources. Station KU02 is located close to the river that can be a source of continuous seismic noise with high amplitudes in the frequency band of 40-80 Hz (Figure 3 (d)). Therefore, for estimation of high-frequency EGF, we pre-filtered the data with the band-pass filter of 1-100 Hz.

## 4.2. Analysis of azimuthal distribution of the noise sources

Classical methods of passive seismic interferometry are based on diffuse field approximation (Wapenaar et al., 2008, 2010). One of the most important conditions for using this approximation is isotropic and homogeneous azimuthal distribution of noise sources (Mulargia, 2012). That is why the second important procedure of data preparation before estimation of EGFs is analysis of the azimuthal distribution of the noise sources during the experiment's period. In our study, we considered two cases. In the case of Pyhäsalmi area, the main sources of high-frequency seismic noise are most probably located inside the mine and in the open pit. Thus, the assumption about isotropic and homogeneous azimuthal distribution is not valid. As shown in Wapenaar et al. (2010), in such cases one cannot assume diffuse field approximation. That is why the measurements of the noise were made along a profile (linear array) consisting of two parts crossing the mine site and oriented EW. However, signals

from other noise sources outside the stationary field area can also present in the wavefield acquired during the data acquisition period. That is why we made additional analysis of azimuthal distribution of noise sources. For calculation of the azimuthal distribution, the well-known methods are frequency-wavenumber (f-k) analysis (Neidell et al., 1971; Douze et al., 1979) and beamforming in the time domain (Rost et al., 2002; Schweizer et al., 2012). The linear configuration of the Pyhäsalmi array

does not allow application of the f-k analysis and beamforming, however. For understanding the directivity of the seismic noise wavefield in different frequency bands, we applied the horizontal-to-vertical ratio rotate method proposed in Nakamura et al. (1989), investigated in Barazza et al. (2009), and implemented into Geopsy software (http://www.geopsy.org).

In our study, we analyse records of seismic noise with duration of 10 min for each hour of records. We applied this procedure to records from stations which are the most distant from the mine and located in both parts of the profile (PLB01 and PLB24).

We have selected two frequency bands (2-5 Hz and 5-10 Hz) for analysis, because they correspond to strong and stable seismic noise, from which it is possible to retrieve surface waves. The result is shown in Figure 4 as a percentage of record time during which the recorded wavefields approached from certain azimuths with respect to the total time of the record. In Figure 4, the azimuth of 0 degree corresponds to the true North and shadowed sectors denote the azimuths to the noise sources. Radial sizes of these sectors are proportional to the relative source-acting time calculated as a percentage of the total measurement time

while angular sizes of the sectors correspond to errors of the azimuth calculation.

In Figure 4 (a, b) one can see strong directivity of the noise wavefields from the East. This proves that the main noise source for the eastern part of the profile and for frequency bands of 2-5 Hz and 5-10 Hz is the mine. Considering the western part of the profile, there is no such clear directivity of the noise wavefields as revealed for the eastern part. One can see near-homogenous azimuthal distribution of the noise sources for azimuths between about 250 and 300 degrees. This could be

explained by location of the profile close to the open pit that occupies a larger area than the underground mine. Because of this, the point-source approximation of noise sources is not valid. From these results we can conclude that if we simply stack all calculated cross-correlation functions for a pair of stations (in particular, in the eastern part of the profile), the final EGF would be biased. Therefore, for estimation of the EGF with minimum bias, we need to apply the advanced method of stacking described above.

In the second case, we considered the KuGB area with low level of high-frequency noise. In order to investigate spatial and azimuthal distribution of the strongest noise sources, we applied the procedure described above to the data of each of small-aperture arrays. The cross-correlation functions were calculated between the central sensor and the other sensors of the array. Figure 5 presents results of the calculations of the azimuths to the strongest seismic noise sources.

In Figure 5, one can see that for the different small-aperture arrays there are also different azimuths to the sources in the

different frequency bands and the directions to the sources depend on frequency. Taking into account the size of our temporary array (aperture of the large array is 3 km and apertures of each small arrays are 0.7 km), we can assume that the sources of high-frequency seismic noise are located at distances larger than 0.7 km, but less than 3 km from the centres of the small-aperture arrays.

## 5 Empirical Green's functions estimation

For estimation of EGFs, it is necessary to apply a procedure for data preparation. This procedure includes several steps, such as spectral whitening, removing parts of records with earthquakes, blasts and missed data. This procedure is applied to the data of both experiments in our study. In the previous parts of our paper, we have demonstrated that the Pyhäsalmi mine is the

source of continuous and strong seismic noise in the frequency band of 1-10 Hz. Therefore, we extract EGFs separately for the eastern and western parts of the profile.

Each part of the profile includes one sensor installed in the closest vicinity of the mine, and we calculated cross-correlation functions between those sensors and each of the other sensors in both parts of the profile. Industrial seismic noise may consist of surface and body waves, because of different types of noise sources.

There are several methods of stacking the cross-correlation functions in the time domain, for example, the root-mean-square method of Shirzad et al. (2014) and the weighted stack by Cheng et al. (2015). We compare the SNR of EGFs estimated by our method for the Pyhäsalmi experiment to the SNR of EGFs estimated by root-mean-square and weighted stacking methods respectively. The SNR was calculated with respect to the surface wave signal seen in EGFs. Results of this comparison are presented in Figure 6.

In Figure 6, one can see that after application of SNR-stacking method we obtained the EGF with the highest signal-to-noise ratio of surface waves, comparing to the other two methods of stacking. This is because we used only cross-correlation functions coherent to each other in our stacks. As one can see from Figure 7, the algorithm selects only about 10% of total number of cross-correlation functions for the final stack. Nevertheless, it does not mean that there are only few cross-correlation functions coherent to each other. It means that after some iterations the signal-to-noise ratio was not increasing any

more by adding new functions to the stack. In other words, the algorithm has found a global maximum of the objective function described in Section 2.

We analysed the apparent velocities obtained from the maxima of each of the cross-correlation functions and the apparent velocities from the cross-correlation functions selected by our algorithm of stacking (Figure 8). This figure shows that most of the retrieved EGFs have group velocities of about 4500 m/s. After applying simple stacking procedure to these cross-

correlation functions, the group velocity of the surface wave part of the resulting EGF is about 4500 m/s. This cannot be true velocity, as it is too high for surface waves propagating in the uppermost bedrock. As can be seen from Figure 8, our SNR-stacking algorithm has selected only EGFs with group velocity of about 3400 m/s. This velocity is close to the minimal value from all group velocities and it is in agreement with group velocities of surface waves and S-wave velocities in the uppermost part of the bedrock in Fennoscandia (Kobranova, 1986; Dortman, 1993; Silvennoinen et al., 2007; Janik et al., 2009; Poli et

al., 2013, etc.). Therefore, after applying our stacking method, we can retrieve EGFs with true group velocity and maximum SNR.

We apply our method of stacking to the cross-correlation functions calculated for the eastern and western parts of the profile for the frequency bands of 2-5 Hz and 5-10 Hz separately. After stacking, we analysed particle-motion diagrams of the waves retrieved from the seismic noise. Figure 9 shows result of stacking and particle motion analysis of EGFs.

In Figure 9, we present only EGFs that probably contain also body waves, because other EGFs, namely those calculated for the western part in the band of 5-10 Hz and for the eastern part in the band of 2-5 Hz contain only surface-wave parts. Figure 9 (a) shows that the seismic noise recorded in the western part of the profile retrieves mainly Rayleigh waves with group velocity of about 3400 m/s. The other wave is marked in Figure 9 (a) as an S-wave because the particle motion diagram corresponds to this type of a wave. Nevertheless, this wave has apparent velocity of 5700 m/s, which is too high. Therefore, we speculate that this can be an artefact and that phase cannot be used for further analysis. In the frequency band of 5-10 Hz, the EGFs calculated for the eastern part of the profile (Figure 10, b) consists of Rayleigh wave. The other arrivals could correspond to one reflected P-wave and three reflected S-waves. Apparent velocities of reflected P-, S1-, S2-, and S3-wave are about 4480 m/s, 3192 m/s, 3261 m/s and 2543 m/s, respectively. Our assumption that these phases may correspond to retrieved body waves is based solely upon comparison of their travel times with the travel times of body waves recorded during previous active source experiment in Pyhäsalmi (Heinonen et al., 2012). Alternatively, the extracted waves may correspond to other phases, for example, to direct waves generated by sources inside the mine. Unfortunately, these assumptions cannot be proved using our data and it would be necessary to use the higher density array for precise phase identification of body waves. In our study, the error in velocities estimation is assumed equal to 0.25 of the wavelength of an extracted signal. The error of the polarization calculation is about 1-3 degrees.

For the KuGB experiment, we calculate cross-correlation functions for each small-aperture array and apply the SNR-stacking algorithm for EGFs evaluation. Cross-correlation functions are calculated between the central sensor and each other sensor of the corresponding small-aperture array. In Figure 10 (b), we present result of EGFs calculation by SNR-stacking method for one of the small-aperture array (SK1-SK8 in Figure 2).

In Figure 10, one can see that after application of simple stacking, there are many implicit maximums in the retrieved EGFs. Due to this, it is not possible to calculate the azimuth to noise sources and apparent seismic wave velocities. However, application of the SNR-stacking allows retrieval of the EGFs with maximums corresponding to surface wave propagating from a virtual source with apparent velocity of about 320-350 m/s. These waves could be Rayleigh wave, or acoustic wave propagating in the air. This assumption is based on the fact that velocity of 350 m/s is close both to the velocity of sound in the atmosphere and to the velocity of surface wave propagating in the shallow quaternary sediments in the uppermost subsurface. For precise determination of the wave type, it would be necessary to have more dense observation network. Nevertheless, using our SNR-stacking algorithm we extracted surface waves from the high-frequency seismic noise. As we noticed in the previous section, the noise sources were distributed stochastically, both in space and in time, and intensity of these noise sources was small. The body waves are not seen in the Figure 10 (b), because a higher density array is necessary for their proper identification.

## 6 Discussion

The classical passive seismic interferometry is based on diffuse-field approximation, because of the equivalence of correlation properties of the multiple-scattering and resulting wavefields. Therefore, it is possible to evaluate EGFs from averaged cross-correlation functions (Campillo et al., 2003). In practice, one needs averaging over long time intervals (more than 1 year)

because of heterogeneous and anisotropic distribution of ambient seismic noise sources during short time intervals (Wapenaar et al., 2010). This is a serious limitation for practical application of passive seismic interferometry as a method of applied geophysics, because it is not always possible to have long-term data acquisition experiments for solving applied problems (mining exploration, microseismic zonation etc.). In such applied problems, the alternative may be to use ballistic waves, not scattered from heterogeneities, but produced by some localized sources of seismic noise (Mulargia, 2012). The major

challenge in this case is retrieving body waves from seismic noise. Recently, some techniques of body-wave extraction were proposed in Almagro Vidal et al. (2014) and Panea et al. (2014). The main idea of these techniques is separating ambient seismic noise into a body-wave part and a surface-wave part. One could expect that combination of these separation methods with our technique of stacking would significantly increase the quality of retrieved body waves. Nevertheless, for this it would be necessary to have the data of dense high-resolution seismic arrays, so making new experiments would be a next step in

development of our technique.

As discussed in Introduction, there are several methods based on weighted-stacking of cross-correlation functions in the time, frequency and time-frequency domains, which allow to increase the quality of extracted EGFs (Shimmel et al., 2011; Cheng et al., 2015; Liu et. al., 2016; Li et. al. 2018, etc.). We showed in previous sections that our time-domain algorithm based on global optimization of the signal-to-noise ratio makes it possible to exclude incoherent cross-correlation functions from

stacking and generally allows obtaining EGFs of even better quality. Of course, signal-to-noise ratio increasing criteria is possible to use in the frequency or in the time-frequency domains, but this would make the algorithm significantly more complicated. It is necessary to remember, however, that this algorithm can be applied because ambient noise sources are generally characterized by relatively wide frequency band. In this case one can expect that increasing signal-to-noise ratio for dominant frequency would result in increase of signal-to-noise ratio of all other frequencies of the signal, as shown in Bensen

et. al. (2007).

The algorithm proposed in this paper has several limitations and drawbacks and hence has potential for improvement. Our technique of stacking allows increasing signal-to-noise ratio, but it has to be applied in combination with array configuration, which allows to keep control over azimuthal distribution of noise sources and to guarantee that the stationary phase condition is satisfied. It can be also envisaged that using mode of noise level distribution instead of the average would make the algorithm

more robust with respect to outliers. Another important limitation is the relationship between the time of seismic wave propagation between neighbouring sensors and the dominant period of the retrieved EGFs. If the time of surface-wave propagation is about one or two periods of this wave, then it would not be possible to separate body and surface waves, similar

to that in seismic experiments with active sources. Moreover, the increase of signal-to-noise ratio of one event, for example, a retrieved surface wave, might lead to decrease of the signal-to-noise ratio of other phases, in particular, body waves.

In certain situations, increasing SNR after addition of a new function to stack is not always reached due to the coherence of the stacked functions. For example, if coda wave part in a cross-correlation function gets smaller, then the SNR increases,

nevertheless stacked functions might not be coherent to each other. The results of testing algorithm with real data demonstrated, however, that the algorithm is robust and works fine with the high-frequency seismic noise acquired in two completely different areas. The results obtained for KuGB area demonstrate that the SNR stacking method might be useful for building up an EGF by stacking ballistic surface wave signals retrieved from the ambient noise. In our study the high quality of surface waves in EGFs was achieved both for brownfield and greenfield exploration areas. Experimental data used in our study is insufficient

to make detailed evaluation how this technique is working with body wave signals, and it will be the subject for our research in the future.

## 7 Conclusion

Results of our study suggest that classical approaches for EGFs evaluation from ambient seismic noise (Campillo, 2003) cannot be considered as a universal tool for extracting high-frequency EGFs. In particular, in quiet areas with low level of

anthropogenic and industrial noise the method would require long registration time because sources of high-frequency wavefield are weak and their distribution is non stationary both in space and time. One of the ways to treat the problem is to use ballistic waves and develop and improve methods for selection of coherent parts of the ambient noise wavefield. Study of azimuthal distribution of ambient noise sources using array techniques is necessary prior to passive seismic experiments, both in greenfield and brownfield exploration areas.

The presented algorithm of cross-correlation functions stacking in a time domain allows to increase significantly signal-to-noise ratio of retrieved EGFs. In our study we demonstrated that under certain conditions the body waves could be extracted from high-frequency industrial seismic noise using the proposed algorithm. This was illustrated with the data collected during passive seismic experiment near the Pyhäsalmi underground mine. Nevertheless, for more detailed testing of possibility of extracting body waves, it would be necessary to analyse the data collected with a higher-density seismic array near mine.

The presented algorithm of stacking makes it possible to extract EGFs from ambient seismic noise with frequencies higher than 1 Hz recorded in quiet areas without strong sources of industrial noise using 2-D seismic arrays. This has been demonstrated by application of our technique to the data collected in the Kuusamo Greenstone belt area that is characterized by the low level of anthropogenic seismic noise and has no industrial sites located nearby.

## 8 Acknowledgements

The study is a part of SEISLAB project funded by the European Regional Development Fund (ERDF), Council of Oulu region (Finland) and Pyhäsalmi Mine Oy and of the ARCEMIS project funded by the KVANTUM Institute of the University of Oulu.

The theoretical part of the study was funded by the Federal Agency of Scientific Organizations the project no. AAAA-A18-118012490072-7 and Research Mobility grant No. 31166 by the Academy of Finland.

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

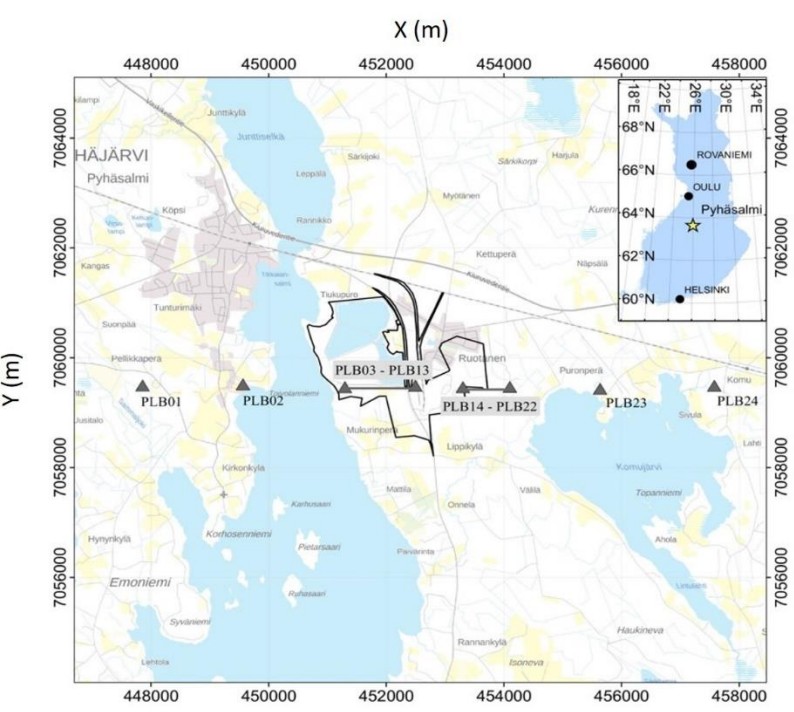

**Figure 1: Map of the experiment near the Pyhäsalmi mine in Universal Transverse Mercator coordinate system with the two parts of the profile (PLB01-PLB13 – west part of profile; PLB14-PLB24 – east part of profile). Black lines are the boarders of the mine and open-pit territories. On locations PLB01, PLB02, PLB03, PLB13, PLB14, PLB22, PLB23, PLB24 both MEMS and Trillium Compact sensors was installed.**

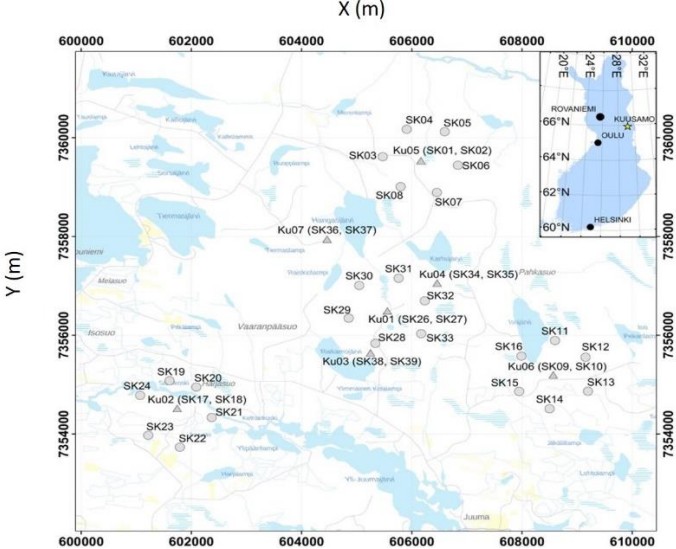

**Figure 2: Configuration of the temporary seismic array on Kuusamo area in Universal Transverse Mercator coordinate system: white triangles – positions of broadband sensors (Trillium compacts); white dots – positions of accelerometers (MEMS).**

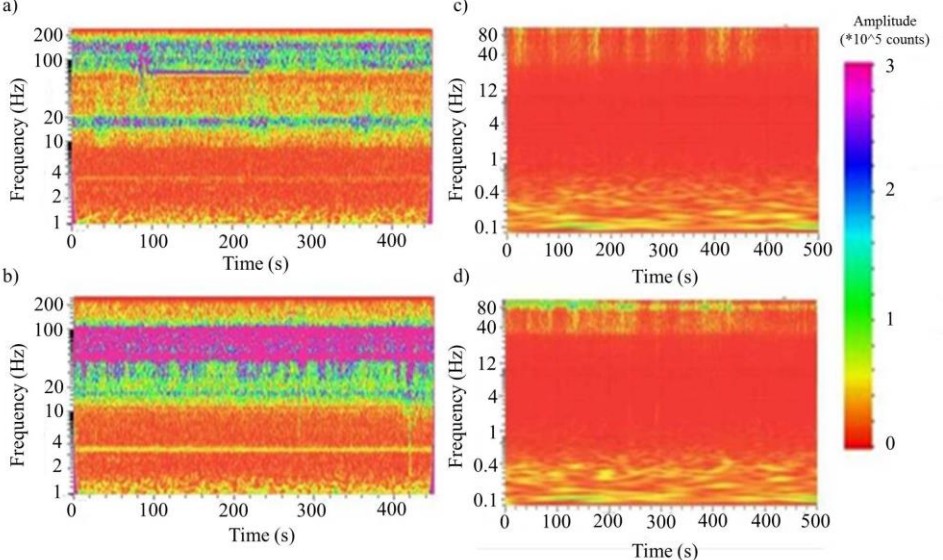

**Figure 3: Result of time-frequency analysis of seismic noise recorded by the sensor a) the most distant from the mine in the Pyhäsalmi experiment, b) closest to the mine in the Pyhäsalmi experiment, c) most distant from a noise source (river) (KU05) in the large-aperture array in Kuusamo experiment, d) KU02 which is closest to the river in the large-aperture array in Kuusamo experiment.**

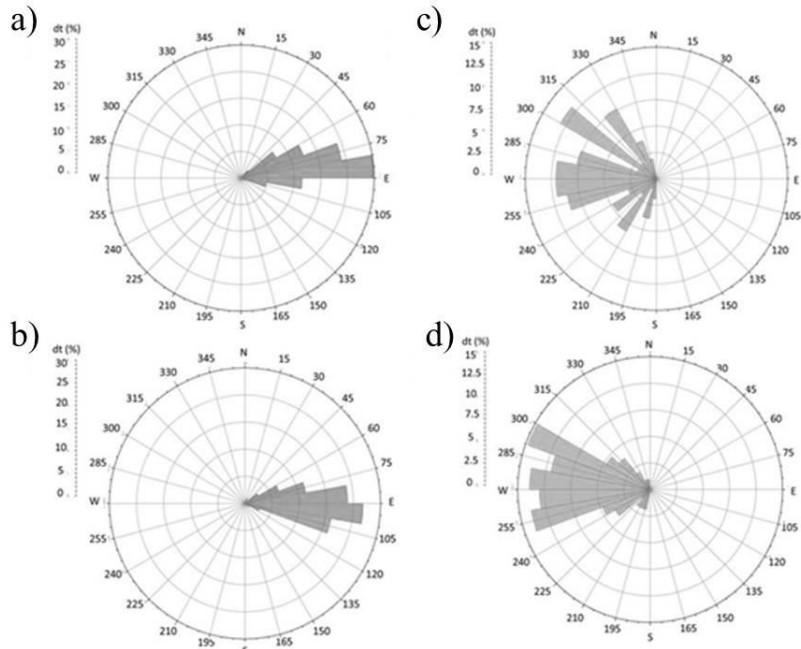

**Figure 4: Result of azimuthal distribution calculation for different frequency bands for the Pyhäsalmi experiment: a) west part of profile, band of 2-5 Hz; b) west part of profile, band of 5-10 Hz; c) east part of profile, band of 2-5 Hz; d) east part of profile, band of 5-10 Hz.**

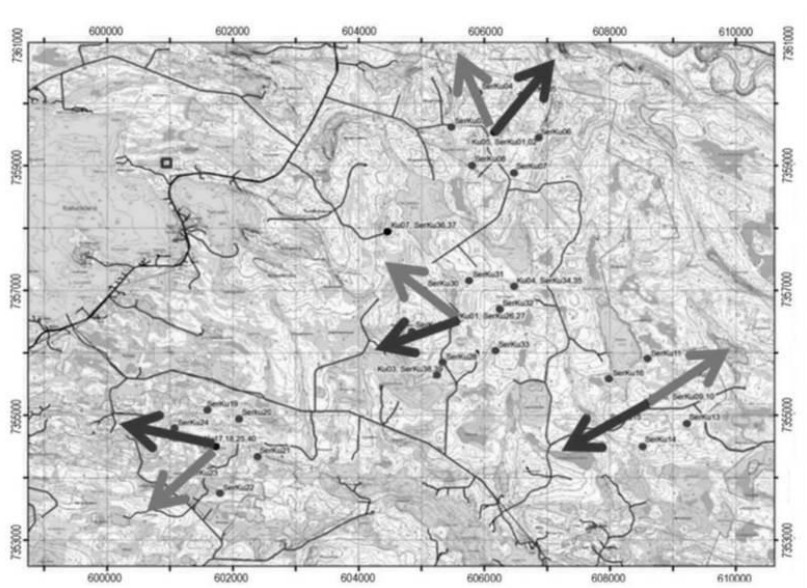

**Figure 5: Azimuths to main noise sources: dots – stations of the temporary seismic array; black arrows show azimuths to noise sources in the frequency band 10-50 Hz; grey arrows show azimuths to noise sources in the frequency band 5-10 Hz.**

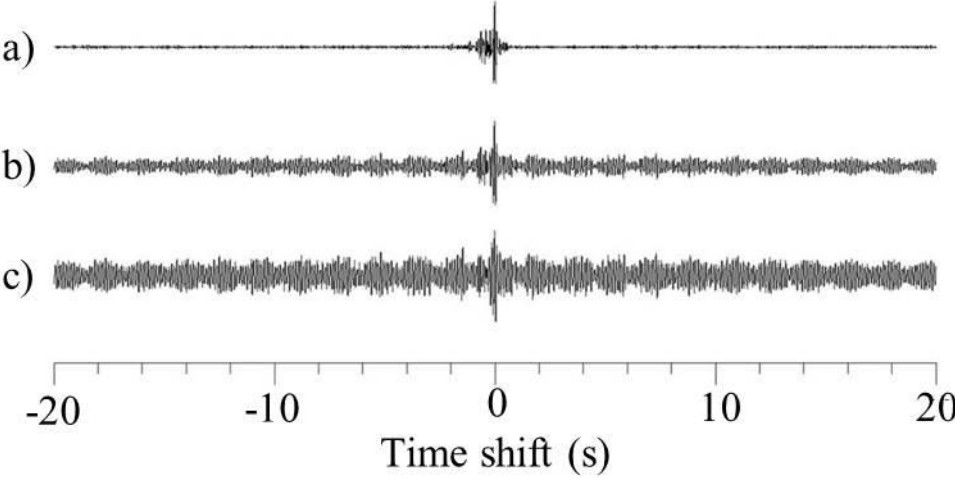

**Figure 6: Final EGF's (vertical components) calculatedby different methods of stacking in the time domain for the frequency band 5-10 Hz: a) SNR-stacking (SNR=40); b) Weight-stacking (SNR=15.6); c) RMS-staking (SNR=10.4).**

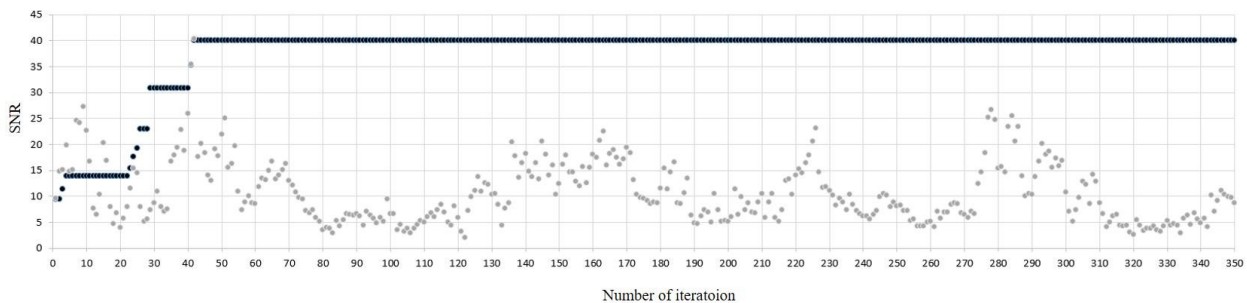

**Figure 7: Build up process of EGF from cross-correlation functions by different methods: black dots – by SNR-stacking, grey dots – by simple stacking.**

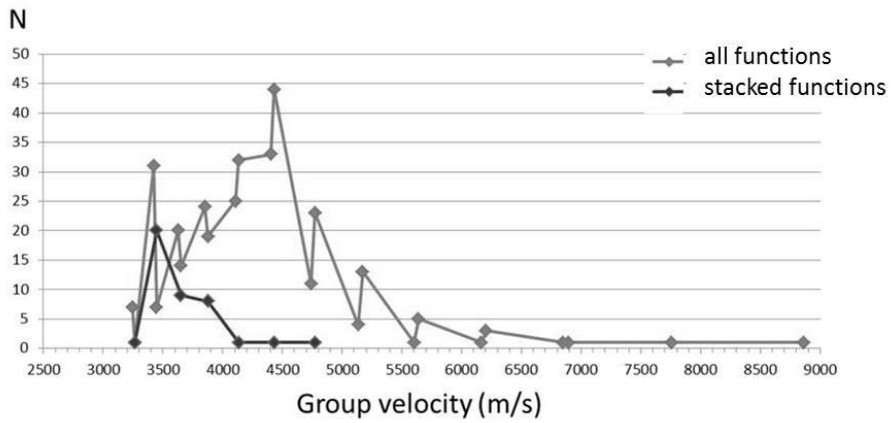

**Figure 8: Distribution of EGF by group velocities for frequencies of 5-10 Hz.**

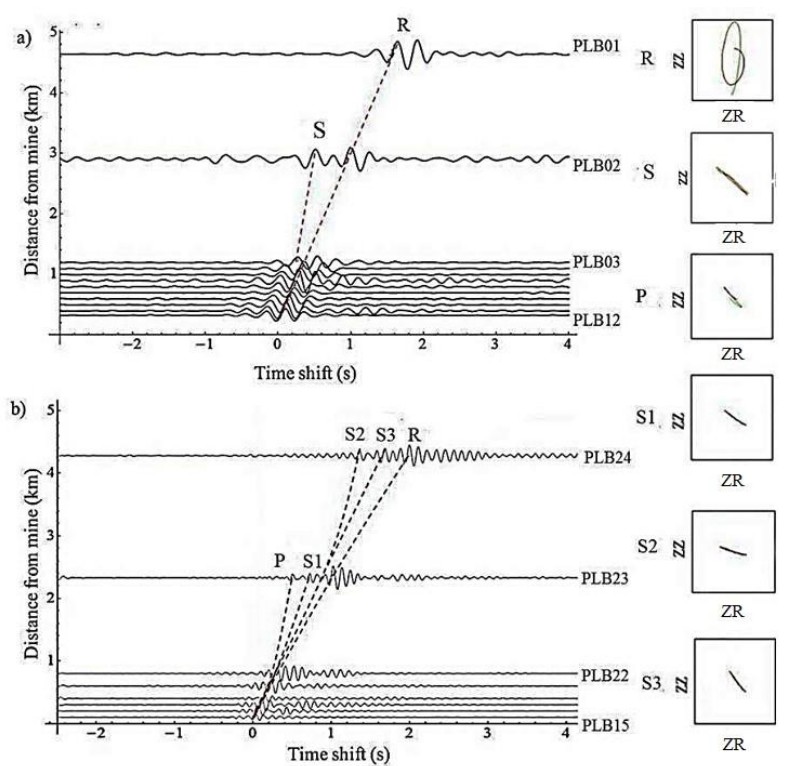

**Figure 9: Result of stacking and particle analysis of EGF, evaluated in Pyhäsalmi experiment: a) western part of the profile in the frequency band 2-5Hz; b) eastern part of the profile in the frequency band 5-10Hz.**

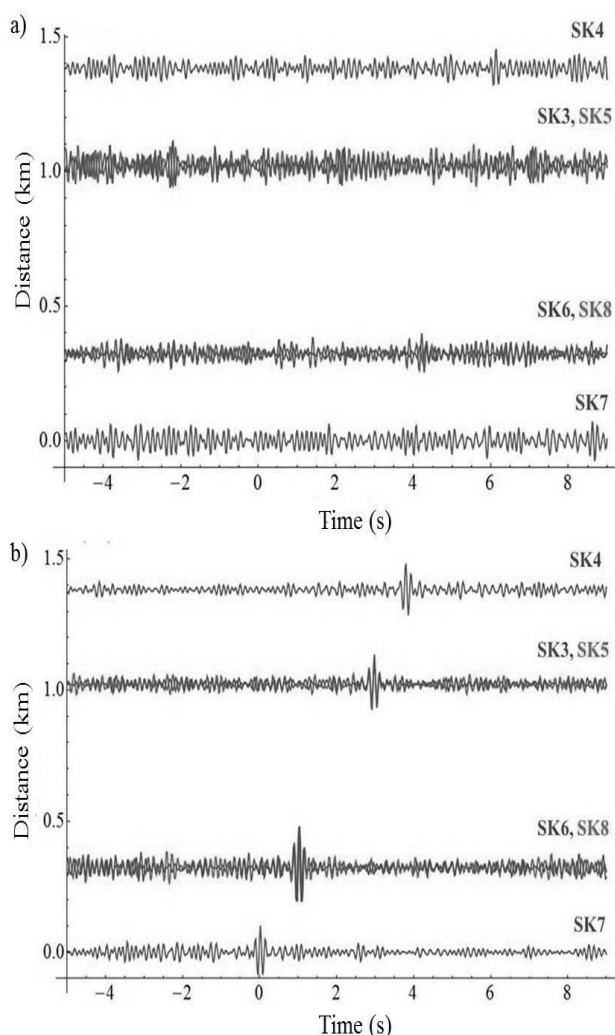

**Figure 10: Empirical Green's functions calculated from records of small-aperture array in Kuusamo experiment in the frequency band of 5-10 Hz and stacked (vertical components): a) by simple stacking method; b) by SNR-stacking method. The EGFs in subplots a) and b) are sorted according to distance from sensor SK7.**