# Peer review of "Improving quality of empirical Greens functions, obtained by crosscorrelation of high-frequency ambient seismic noise"

_Solid Earth, 2019_

## Referee Comment (RC1) · Anonymous Referee #1 · 5 Mar 2019

This paper proposes an improvement to the method of Green's function retrieval from ambient noise by cross-correlation. A specific stacking method is proposed which discards partial correlation results that are not coherent with the average correlation result. After applying an iterative procedure, a correlation function is obtained with a higher signal-to-noise ratio than the ones obtained by other stacking methods. The method is illustrated with two preliminary field data examples. The authors discuss the advantages and limitations of the method.

This reviewer is familiar with the theory of Green's function retrieval but does not have a

broad overview of the many processing methods that have been developed. Therefore it is difficult to judge the originality of the proposed method. I recommend that the paper be reviewed at least by one additional reviewer, who is more experienced with the practical aspects of Green's function retrieval.

Assuming the proposed method is original, I recommend publication after moderate revision, taking the following comments into account:

- I wonder why the authors call their method "signal-to-noise ratio (SNR) stacking". Aren't all stacking methods aiming to improve the SNR? The proposed method stands out because it discards incoherent correlation results. Please consider a new name, which better matches the specific aspects of the proposed method. For example: "Coherent stacking"? "Coherent cross-correlation stacking"?

- On page 2 the authors mention that they want to use high-frequency surface waves to extract information about deep structures. This sounds as a contradiction. Surface waves do not penetrate deep into the subsurface, and using high frequencies makes it even more difficult to reach deep structures. Please be quantitative about the depths that need to be reached.

- Page 3, line 2. The introduction of $\Delta t_e$ via the inequality is confusing. Is $\Delta t_e$ the time-lag interval, or is the inequality $-t_{ds} < \Delta t_e < t_{ds}$ the time-lag interval (as actually stated in line 2)? If $\Delta t_e$ is the time-lag interval (as stated in line 7), what does it mean that it can take a negative value (as stated in line 2)? Please explain.

- Explain abbreviations, such as MEMS and BB sensors.

- Mention the area of the experiments in all figure captions (Fig1: Pyhäsalmi mine area, Fig2: Kuusamo Greenstone Belt area, etc.).

[Figure]

- Figure 6a: I am surprised that the time-shift of the peak appears almost at t=0. Why don't you show a more representative example with a time-shifted peak, corresponding to well-separated receivers?

- Figures 6b and 6c: I think these figures (or the corresponding captions) should be interchanged: the SNR in 6b looks better than that in 6c, but the captions say the opposite.

- Figure 7. The SNR of the proposed method converges to 40. However, according to the caption of fig 6a the SNR equals 71. Please explain. Are these different experiments?

- Figs 9 and 10 show only some preliminary results of the method for both regions. These figures show that Green's functions can be retrieved and the derived velocities seem to be in agreement with earlier derived results. I would have liked to see more discussion on what can be done with these results (or do we need more data before useful inferences about the area of investigation can drawn?)

- Last but not least, the paper needs significant language editing!
* * *

---

## Referee Comment (RC2) · Anonymous Referee #2 · 2 Apr 2019

**Improving quality of empirical Greens functions, obtained by cross-correlation of high-frequency ambient seismic noise**

**General overview**

A main problem in exploration geophysics applications using anthropogenic sources of seismic ambient noise often is its far from ideal distribution that hinders the extraction of empirical Green's function using methods conventionally used at much larger scales using natural sources, for example, in seismology. The authors introduce a method that seek for a subset of interstation correlations that maximize the signal-to-noise ratio (SNR) after stacking to promote converge to the empirical Green's function. Overall the manuscript is interesting but the English usage has to improve and many places need for greater clarity. I just have a few comments.

**Main comments**

- Pg. 2: The introduction on the stacking methods is confusing. I would distinguish methods that weight correlations according to the SNR of each correlation

(Cheng et al., 2015) or that stack only correlations with high or low coherence (Boué et al., 2014) from methods that weights signal coefficients in a transformed domain after a linear stack, such as the time-frequency phase weighted stack (Baig et al., 2009; Schimmel et al., 2011; Li et al., 2018) or the time-scale phase weighted stack (Ventosa et al., 2017). A clear separation between these methods can help the reader to place your method in its proper context.

- Pg. 2, line 10 and 14: Li et al. (2017) should be Li et al. (2018).

- Pg. 2, line 25-26: Can you give detailed information on the pre-processing and the cross-correlation function you apply?

- Pg. 2, line 28-30 and Pg. 3, line 11: These sentences are misleading. Interstation correlation functions do not always give an empirical Green's function. They converge to an empirical Green's function when the distribution of source is fairly well distributed. Hence, the importance of the pre-processing, correlation, stacking methods, and potentially the method you introduce, to seek a good balance of sources.

- Pg. 3, eq. (1): This estimation of SNR is fine when the strongest signals arrive on the expected time lags (from $-t_{ds}$ to $t_{ds}$) and you have no signal outside. Have you considered using more robust estimators of the noise level such as median absolute deviation (MAD). What happens when signals are too weak to be observed in a cross-correlation function but arise after stacking?

- Pg. 4, lines 14-21: This paragraph is not clear. If I understood what you mean, you need to know seismic velocity in order to measure the azimuth of the strongest source; however, seismic velocity structure is often what we seek in most applications. In addition, you mention that a 2-D array is necessary. Can you further explain how you use it to estimate the azimuth distribution of sources.

- Pg. 5, around Fig. 1 & 2: I would personally emphasize in these figures which stations use MEMS and which Trillium Compact.

- Pg. 8, line 1: An extra sentence is needed here to explain how you locate high-frequency noise sources at distances from about 0.7 to 3 km from the center of the arrays.

- Pg. 8, line 17-22: Which is the portion of correlations that conventionally build up the final stack?

**References**

Boué, P., Poli, P., Campillo, M. and P. Roux (2014). Reverberations, coda waves and ambient noise: Correlations at the global scale and re-trieval of the deep phases, Earth Planet. Sci. Lett., 391, 137–145, doi:10.1016/j.epsl.2014.01.047

Baig, A., Campillo, M. and F. Brenguier, F. (2009). Denoising seismic noise cross correlations, J. Geophys. Res., 114(B8), 2156–2202, doi:10.1029/2008JB006085

Ventosa, S., Schimmel, M. and E. Stutzmann (2017). Extracting surface waves, hum and normal modes: time-scale phase-weighted stack and beyond, Geophys. J. Int., 211(1), 30–44, doi:10.1093/gji/ggx284

---

## Short Comment (SC1) · 2 Apr 2019

Please see below the comment to the manuscript about method for retrieval of Green's function (GF) with high S/N ratio in selected time window. The review is submitted as Short Comment to online discussion: https://www.solid-earth-discuss.net/se-2019-29/.

This post is encouraged by one of the comments of the Anonymous Refree #1 suggesting to focus on the originality of proposed ambient-noise processing technique. In this paper authors propose a method to retrieve improved version of Green's function between receiver pairs and apply it on two different datasets. The paper is enjoyable to read and seems like a great case study.

[Figure]

The method is based on rejecting cross-correlation functions which after stacking do not increase the S/N ratio in the time window related to arrivals of the desired phases. The S/N ratio in this method is calculated according to equation 1 (Page 3), and generally is obtained by dividing the maximum amplitude in time interval of expected arrival by the summed amplitudes in the remaining part of CCF. If adding the CCF does not increase the S/N ratio, then it is rejected.

Generally all methods basing on S/N criteria are robust and effective, and they are commonly used as part of ambient-noise processing workflows. The main issue of 'S/N ratio stacking' proposed here is that the method seems to be not novel. To give some examples please see the papers by Olivier et al. (2015) and Nakata et al. (2015). Both papers describe the process of extracting body-waves form ambient noise and both apply S/N ratio based method as one of the steps in processing workflow.

Olivier et al. (2015) designs the selective stacking algorithm for enhancing the S-wave arrivals recorded with array of receivers in the underground mine. In their method the root-mean-square value (RMS) of the signal in the lag-time window of the correlation function around the expected arrival times of the S-waves is divided by the RMS of the signal in the time window of coda waves. It is practically the same method, just instead of maximum amplitude authors use rms.

Nakata et al. (2015) as part of his ambient-noise processing designs two different S/N ratio based techniques. First one is more elaborate, so please see the mentioned publication. The second one is (direct citation from paper): "To confirm that we can successfully isolate the traces with strong body wave energy with the second correlation, we compute SNR, which is defined as the average RMS amplitudes between 1.3 and 1.9 s divided by the average RMS amplitudes between 0.0 and 4.5 s." – again please note the striking similarity of the method.

It is important to note that the two above techniques were just one step of the more elaborated processing workflows, and both of the mentioned papers included also extensive synthetic tests and applications of tomography.

Second part of comment is related to the line 15 (Page 2) in the discussion manuscript where authors provide their definition of 'coherent' term. According to this definition the two EGFs are coherent if their maxima fall in the same time window (appear at the same time-lag). While, this definition of coherence is comfortable in terms of improving Green's function it might not necessarily be correct for the field applications.

In lines 25-30 (Page 2) Authors argue that stacking only EGFs with which increase S/N ratio given in equation 1, does automatically increase the coherency. This is true, but only for the specific definition of coherency given in this manuscript, which however does not relate to retrieval of correctly estimated Green's function, which needs source in the stationary phase areas. In line 10 (page 2) authors indeed comment that its important to use systems which allows to estimate the azimuth distribution of noise sources (to increase a chance of capturing the sources in stationary phase areas), yet this comment does not suffice to make a method feasible for improved processing, as usually the exact distribution of sources is not known. In such cases, specific methods can be used for estimation these azimuths (like beamforming etc.), yet when this directional analysis is already done, then it is enough just to stack these sources. After this, any measure of the increase of amplitude in expected time window becomes trivial task.

Generally it is reasonable to measure the EGF using coherency because it will, in ideal situation, selectively correct virtual traces which contribute to the stack. However, using S/N ratio in selected time windows might not be necessarily correct, as the source we are stacking might be located in non-stationary phase areas. In other words, the maximum amplitude we eventually get, may not mean we stack sources related to the stationary phases (which depends on the source-receiver configuration).

Second issue related to possibly biased coherency improvement is related to the division in equation 1. The coherency improvement is theoretically assured if S/N ratio

calculated from equation 1 is increasing. This might not be necessarily true, e.g., if coda wave part gets smaller (the denominator in equation 1) the S/N also increases, and again it means that source contributing to desired time-windows might be not related to the stationary region.

Thanks for reading and looking forward to your reply.

Kind regards, Michal Chamarczuk

---

## Author Comment (AC1) · 1 May 2019

Referee: This paper proposes an improvement to the method of Green's function retrieval from ambient noise by cross-correlation. A specific stacking method is proposed which discards partial correlation results that are not coherent with the average correlation result. After applying an iterative procedure, a correlation function is obtained with a higher signal-to-noise ratio than the ones obtained by other stacking methods.

The method is illustrated with two preliminary field data examples. The authors discuss the advantages and limitations of the method. This reviewer is familiar with the theory of Green's function retrieval but does not have a broad overview of the many processing methods that have been developed. Therefore it is difficult to judge the originality of the proposed method. I recommend that the paper be reviewed at least by one additional reviewer, who is more experienced with the practical aspects of Green's function retrieval. Assuming the proposed method is original, I recommend publication after moderate revision, taking the following comments into account: • I wonder why the authors call their method "signal-to-noise ratio (SNR) stacking". Aren't all stacking methods aiming to improve the SNR? The proposed method stands out because it discards incoherent correlation results. Please consider a new name, which better matches the specific aspects of the proposed method. For example: "Coherent stacking"? "Coherent cross-correlation stacking"?

Authors: We agree that all stacking methods aim to increase SNR of evaluated EGF. Nevertheless, we call the method "SNR-stacking", because of using "signal-to-noise ratio" as a parameter that is optimized in our suggested algorithm. The methods perform global optimization search by retrieving EGF with the highest SNR. Using other terms like "coherence", in our opinion, may mislead readers because we use this term only in order to shorten the description of the method. Our definition of the term "coherence" is defined in the Introduction part of the manuscript.

Referee: • On page 2 the authors mention that they want to use high-frequency surface waves to extract information about deep structures. This sounds as a contradiction. Surface waves do not penetrate deep into the subsurface, and using high frequencies makes it even more difficult to reach deep structures. Please be quantitative about the depths that need to be reached.

Authors: The sentence with the description of the depth of investigation has been corrected as proposed by Referee.

Referee: • Page 3, line 2. The introduction of _te via the inequality is confusing. Is _te the time-lag interval, or is the inequality ôĂĂĂtds < _te < tds the time-lag interval (as actually stated in line 2)? If _te is the time-lag interval (as stated in line 7), what does it mean that it can take a negative value (as stated in line 2)? Please explain.

Authors: This was a mistake in the definition. The _te is time lag, not time lag interval. An additional explanation has been added to the text. Our algorithm is based on global optimization trying to optimize the SNR and we are calculating the SNR as a function of time lag that is variable and also other variables such as initial function number etc.) with the expected signal. In most cases, we do not know the azimuthal distribution of noise sources. That is why we need to consider both casual (positive time lags) and acasual (negative time lags) parts of crosscorrelation functions. In this case, we use the time interval with zero point at _te with a width of two periods of expected signal.

Referee: • Explain abbreviations, such as MEMS and BB sensors.

Authors: MEMS – microelectromechanical system. BB – broadband. The explanation has been added to the text.

Referee: • Mention the area of the experiments in all figure captions (Fig1: Pyhäsalmi mine area, Fig2: Kuusamo Greenstone Belt area, etc.).

Authors: Figure captions are corrected as proposed.

Referee: • Figure 6a: I am surprised that the time-shift of the peak appears almost at t=0. Why don't you show a more representative example with a time-shifted peak, corresponding to well-separated receivers?

Authors: On figure 6 we show differences in signal-to-noise ratio for EGFs, obtained by different methods of stacking. In this case, the time lag, which corresponds to signal, is small compared to the length of noise wavetrain, and on the figure it looks like zero-lag. Nevertheless, for illustration of the quality of EGF obtained, it is necessary to show the whole time interval that was used for calculation of noise level. It seems for us, that for

visualisation of signal-to-noise ratio improvement it is better to use a large time window, in which the difference between noise and signal is seen better.

Referee: • Figures 6b and 6c: I think these figures (or the corresponding captions) should be interchanged: the SNR in 6b looks better than that in 6c, but the captions say the opposite.

Authors: This was a typo and it has been corrected.

Referee: • Figure 7. The SNR of the proposed method converges to 40. However, according to the caption of fig 6a the SNR equals 71. Please explain. Are these different experiments?

Authors: This was a typo and it has been corrected.

Referee: • Figs 9 and 10 show only some preliminary results of the method for both regions. These figures show that Green's functions can be retrieved and the derived velocities seem to be in agreement with earlier derived results. I would have liked to see more discussion on what can be done with these results (or do we need more data before useful inferences about the area of investigation can drawn?)

Authors: Extracted empirical Greens functions can be processed by the same techniques as a signal from a controlled source. Further processing of the signal (EGF in our case) is simpler if the signal-to-noise ratio is relatively higher. The goal of our paper is to describe a method for improving EGFs quality and its possibilities. We plan to use the method with the data of other experiments, with a larger number of sensors.

Referee: • Last but not least, the paper needs significant language editing! Authors: We are very thankful to the reviewer for this comment and additional text editing was done.

---

## Author Comment (AC2) · 1 May 2019

Michał Chamarczuk mchamarczuk@igf.edu.pl

Referee: Please see below the comment to the manuscript about method for retrieval of Green's function (GF) with high S/N ratio in selected time window. This post is encouraged by one of the comments of the Anonymous Refree #1 suggesting to focus on the originality of proposed ambient-noise processing technique. In this paper authors propose a method to retrieve improved version of Green's function between receiver

pairs and apply it on two different datasets. The paper is enjoyable to read and seems like a great case study. The method is based on rejecting cross-correlation functions which after stacking do not increase the S/N ratio in the time window related to arrivals of the desired phases. The S/N ratio in this method is calculated according to equation 1 (Page 3), and generally is obtained by dividing the maximum amplitude in time interval of expected arrival by the summed amplitudes in the remaining part of CCF. If adding the CCF does not increase the S/N ratio, then it is rejected. Generally all methods basing on S/N criteria are robust and effective, and they are commonly used as part of ambient-noise processing workflows. The main issue of 'S/N ratio stacking' proposed here is that the method seems to be not novel. To give some examples please see the papers by Olivier et al. (2015) and Nakata et al. (2015). Both papers describe the process of extracting body-waves form ambient noise and both apply S/N ratio based method as one of the steps in processing workflow. Olivier et al. (2015) designs the selective stacking algorithm for enhancing the S-wave arrivals recorded with array of receivers in the underground mine. In their method the root-mean-square value (RMS) of the signal in the lag-time window of the correlation function around the expected arrival times of the S-waves is divided by the RMS of the signal in the time window of coda waves. It is practically the same method, just instead of maximum amplitude authors use rms. Nakata et al. (2015) as part of his ambient-noise processing designs two different S/N ratio based techniques. First one is more elaborate, so please see the mentioned publication. The second one is (direct citation from paper): "To confirm that we can successfully isolate the traces with strong body wave energy with the second correlation, we compute SNR, which is defined as the average RMS amplitudes between 1.3 and 1.9 s divided by the average RMS amplitudes between 0.0 and 4.5 s." – again please note the striking similarity of the method. It is important to note that the two above techniques were just one step of the more elaborated processing workflows, and both of the mentioned papers included also extensive synthetic tests and applications of tomography.

Authors: 1) The novelty of our technique compared to the other techniques mentioned

by the reviewer is that we applied global optimization algorithm to objective function (SNR of EGF in our case) for evaluation of the best solution (EGF of the highest quality). In our proposed algorithm, we calculate SNR as a function of several parameters (time lags with an expected signal, initial time windows number etc., see the algorithm description). A parametrisation of the global optimization problem is based on the a-priori information and generally, is problem-dependent. After this, the algorithm finds the best solution corresponding to the global maximum of SNR function.

2) The other important feature of our algorithm is that the signal-to-noise ratio is estimated in the time-domain and hence the objective function in global optimization problem is sensitive to variations of not only the RMS, but also to other parameters. For example, changing the azimuth to noise source will shift the position of the signal maximum in the time window considered. In this case, the RMS for this window may be the same, but the position of the maximum will be shifted. Therefore, our algorithm will reject this function, while algorithms based on RMS would not. It is true that our method is using the ideas proposed by other authors, and we cited all these studies in our paper. However, we developed the original method of signal-to-noise ratio optimisation, which is more sensitive, because we use the maximum of CCF instead of RMS. We compared results obtained by several methods (RMS-based stacking and weight stacking (figure 6)) and found out, that our proposed technique allows obtaining EGFs of better quality. Moreover, we advanced the algorithm of stacking by using of global optimization of SNR, which makes results more robust and independent on initial cross-correlation function. Suggested papers have been cited in the text.

Referee: Second part of comment is related to the line 15 (Page 2) in the discussion manuscript where authors provide their definition of 'coherent' term. According to this definition the two EGFs are coherent if their maxima fall in the same time window (appear at the same time-lag). While, this definition of coherence is comfortable in terms of improving Green's function it might not necessarily be correct for the field applications.

Authors: We used the term "coherence" in order to simplify the description of the method and we explained in what sense it is used. We think that our definition is close to the standard definition of this term in physics, but in our case, a wave is a cross-correlation function. In our case, increasing of SNR after stacking of two cross-correlation functions is the same as a result of interference of two waves which are coherent to each other. There are also some differences from the standard physical definition of coherence. For example, we use time lags with maximums instead of phase differences.

Referee: In lines 25-30 (Page 2) Authors argue that stacking only EGFs with which increase S/N ratio given in equation 1, does automatically increase the coherency. This is true, but only for the specific definition of coherency given in this manuscript, which however does not relate to the retrieval of correctly estimated Green's function, which needs source in the stationary phase areas.

Authors: We agree with this comment, but we explained in our paper in what sense we use the term "coherence". Our proposed technique allows increasing signal-to-noise ratio, but it does not guarantee to estimate of the true EGF, because the source may be located outside the stationary phase area. Nevertheless, using our technique together with the array analysis techniques, which allow estimating azimuth to the noise source, makes it possible to evaluate EGF of high quality. The discussion has been added to the Conclusion part of the manuscript.

Referee: In line 10 (page 2) authors indeed comment that its important to use systems which allows to estimate the azimuth distribution of noise sources (to increase a chance of capturing the sources in stationary phase areas), yet this comment does not suffice to make a method feasible for improved processing, as usually the exact distribution of sources is not known. In such cases, specific methods can be used for estimation these azimuths (like beamforming etc.), yet when this directional analysis is already done, then it is enough just to stack these sources. After this, any measure of the increase of amplitude in expected time window becomes trivial task.

Authors: In our paper, we consider two cases that are of practical importance for geo-physical explorations: the first one is brownfield exploration, in which position of the dominating noise source is relatively well known (e.g. mine) and the second one is greenfield exploration, in which we have no any a-priory information about spatial and temporal distribution of noise sources. In the first case, after directional analysis of noise sources, the EGF evaluation is an easy task if one of the following conditions is satisfied: 1) the azimuthal distribution is homogeneous; 2) there are sources located in some limited area and producing noise of high energy. However, if the noise sources are stochastically distributed both in time and in space and are weak, then using sim-ple stacking for extraction of EGF is not a guarantee of a good result, even if one can estimate the azimuthal distribution of noise sources, and evaluation of EGF become not a trivial task. We demonstrated this by our Kuusamo experiment (see fig. 10).

Referee: Generally, it is reasonable to measure the EGF using coherency because it will, in ideal situation, selectively correct virtual traces, which contribute to the stack. However, using S/N ratio in selected time windows might not be necessarily correct, as the source we are stacking might be located in non-stationary phase areas. In other words, the maximum amplitude we eventually get, may not mean we stack sources related to the stationary phases (which depends on the source-receiver configuration).

Authors: This issue is partially solved by using global optimization of SNR in our pro-posed algorithm (see also our reply to the comment above).

Referee: Second issue related to possibly biased coherency improvement is related to the division in equation 1. The coherency improvement is theoretically assured if S/N ratio calculated from equation 1 is increasing. This might not be necessarily true, e.g., if coda wave part gets smaller (the denominator in equation 1) the S/N also increases, and again it means that source contributing to desired time-windows might be not related to the stationary region.

Authors: This is one of the possible problems of the method. The correspondent discussion has been added to the text. But in our paper, we considered two real data cases that are of practical importance for geophysical exploration, and we demonstrated that method is working.

Thanks for reading and looking forward to your reply. Kind regards, Michal Chamarczuk

---

## Author Comment (AC3) · 1 May 2019

Improving quality of empirical Greens functions, obtained by cross-correlation of high-frequency ambient seismic noise

General overview A main problem in exploration geophysics applications using anthropogenic sources of seismic ambient noise often is its far from ideal distribution that hinders the extraction of empirical Green's function using methods conventionally used at much larger scales using natural sources, for example, in seismology. The authors

introduce a method that seek for a subset of interstation correlations that maximize the signal to-noise ratio (SNR) after stacking to promote converge to the empirical Green's function. Overall the manuscript is interesting but the English usage has to improve and many places need for greater clarity. I just have a few comments.

Main comments

Referee: • Pg. 2: The introduction on the stacking methods is confusing. I would distinguish methods that weight correlations according to the SNR of each correlation (Cheng et al., 2015) or that stack only correlations with high or low coherence (Boué et al., 2014) from methods that weights signal coefficients in a transformed domain after a linear stack, such as the time-frequency phase weighted stack (Baig et al., 2009; Schimmel et al., 2011; Li et al., 2018) or the time-scale phase weighted stack (Ventosa et al., 2017). A clear separation between these methods can help the reader to place your method in its proper context.

Authors: Thanks a lot for more clear formulation of the sentence. Correspondent text in the manuscript has been changed.

Referee: Pg. 2, line 10 and 14: Li et al. (2017) should be Li et al. (2018).

Authors: This work published in 2017: Li, G., Niu, F., Yang, Y., & Xie, J. (2017). An investigation of time–frequency domain phase-weighted stacking and its application to phase-velocity extraction from ambient noise's empirical Green's functions. Geophysical Journal International, 212(2), 1143-1156.

Referee: •Pg. 2, line 25-26: Can you give detailed information on the pre-processing and the cross-correlation function you apply?

Authors: There is a number of studies devoted to calculating of EGF from ambient seismic noise. In our study, we used the preprocessing routine described in details in Benson et al. (2007), Poli et al. (2012, 2013). These procedures now are the "standard" pre-processing procedures in passive seismic interferometry. That is why we did

not concentrate in our paper on these details, but only refer to the papers mentioned above.

Referee: • Pg. 2, line 28-30 and Pg. 3, line 11: These sentences are misleading. Interstation correlation functions do not always give an empirical Green's function. They converge to an empirical Green's function when the distribution of source is fairly well distributed. Hence, the importance of the pre-processing, correlation, stacking methods, and potentially the method you introduce, to seek a good balance of sources.

Authors: We put a corrected, more clear explanation into the text.

Referee: • Pg. 3, eq. (1): This estimation of SNR is fine when the strongest signals arrive on the expected time lags (from ôĂĂĂtds to tds) and you have no signal outside. Have you considered using more robust estimators of the noise level such as median absolute deviation (MAD). What happens when signals are too weak to be observed in a cross-correlation function but arise after stacking?

Authors: This estimation is out of the scope of our paper. Signal-to-noise ratio could be estimated by several methods. Of course, the results depend on the choice of method for this calculation. The main thing that we were going to demonstrate in our paper is our technique that is using the stacking method together with the global optimization of the signal-to-noise ratio. Analysis of different methods of SNR calculation is a task for our further studies, and we agree that there is a potential for further improvement of the method.

Referee: • Pg. 4, lines 14-21: This paragraph is not clear. If I understood what you mean, you need to know seismic velocity in order to measure the azimuth of the strongest source; however, seismic velocity structure is often what we seek in most applications. In addition, you mention that a 2-D array is necessary. Can you further explain how you use it to estimate the azimuth distribution of sources.

Authors: Estimation of time lags intervals with expected signals is calculated according
to the plane wave condition as in beamforming. Limits of velocities are selected according to a priory information about the studied medium. After this, such parameters as the SNR, apparent velocity and azimuths are optimized. Therefore, we can estimate probable values of azimuth and velocity. The explanation has been added to the text.

Referee:  c Pg. 5, around Fig. 1 & 2: I would personally emphasize in these figures which stations use MEMS and which Trillium Compact.

Authors: Figure caption has been corrected.

Referee:  c Pg. 8, line 1: An extra sentence is needed here to explain how you locate highfrequency noise sources at distances from about 0.7 to 3 km from the center of the arrays.

Authors: As we apply standard array methods for location of noise sources, we assumed these values from apertures of these arrays. The additional sentence has been added to the text.

Referee:  c Pg. 8, line 17-22: Which is the portion of correlations that conventionally build up the final stack?

Authors: It is difficult to estimate because it is strongly dependent on features of the noise wavefield. In two cases considered in our study, the number of cross-correlation functions used in the final stack varies from about 8 to 35% of the total number of calculated functions.

References Boué, P., Poli, P., Campillo, M. and P. Roux (2014). Reverberations, coda waves and ambient noise: Correlations at the global scale and retrieval of the deep phases, Earth Planet. Sci. Lett., 391, 137–145, doi:10.1016/j.epsl.2014.01.047 Baig, A., Campillo, M. and F. Brenguier, F. (2009). Denoising seismic noise cross correlations, J. Geophys. Res., 114(B8), 2156–2202, doi:10.1029/2008JB006085 Ventosa, S., Schimmel, M. and E. Stutzmann (2017). Extracting surface waves, hum and normal modes: time-scale phase-weighted stack and beyond, Geophys. J. Int., 211(1),

30–44, doi:10.1093/gji/ggx284

---

## Author Response (AR2)

**Authors response to anonymous referee 2**

Suggestions for revision or reasons for rejection

**Referee:**

Thanks for the last corrections and additions. The manuscript and the English usage has improved a bit; however, there is still much room for improvements and main issues have not been solved. In particular, the introduction has to improve a lot to allow the reader to put your method in context and to clearly know the similarities and the differences with existing alternative methods. In addition to field data examples, synthetic examples and comparisons with conventionally used methods are often essential to illustrate the key contributions.

**Authors:**

Thanks a lot for the pointing to some typos and misleading in the text.

The main advantage of our method is using of global optimization algorithm for obtaining the solution with the best "signal-to-noise ratio". The global optimization approach is widely used for solution of inversion problems in geophysics. Moreover, it is also used in the recent applications of various artificial intelligence techniques (machine learning, deep learning) for processing of geophysical data. In our case, a SNR, calculated on each iteration, is an objective function in some parameter space (see Section 2 for description of model parameters) that is optimized. Global optimization approach ensures that the algorithm finds not some local, but global optimum for SNR. It is the main difference of our and existing approaches, described in numerous papers.

The corresponding text have been added to the Abstract and Introduction.

**Referee:**

MAIN COMMENTS

The third paragraph of the introduction (Pg. 2, lines 9-27) has several errors and some parts are misleading.

• Line 16: The time-frequency phase weighted stack (tf-PWS) was introduced by Baig et al. (2009) and Schimmel et al. (2011). Li et al. (2017) study the errors that some inverse S-transforms may introduce on subsequent phase-velocity measurements.

**Authors:**

This part of the introduction has been corrected. In our text we refer to all these authors.

**Referee:**

• I have to insist Li et al. (2017) should be Li et al. (2018), doi: 10.1093/gji/ggx448 . Although Advance Access publication dates from October 2017, the official publication date is from February 2018. I will use Li et al. (2017) below to avoid confusion.

**Authors:**

Reference to this paper has been changed as proposed.

**Referee:**

• As its name indicates, the time-frequency phase weighted stack studied by Li et al. (2017) is using stacking in the time-frequency domain not the frequency domain.

**Authors:**

These typos have been corrected.

**Referee:**

• Line 17: The time-scale phase weighted stacking from Ventosa et al. (2017) is not using stacking in the domain. It works on the time-frequency domain, as the tf-PWS, using the wavelet transform instead of the S-transform.

**Authors:**

This part of text has been corrected.

**Referee:**

• Line 17-19: This sentence has not much sense. Coherence can be written as a function signal-to-noise ratio (SNR). In order to improve SNR, the tf-PWS methods focus on weighting the components of the cross-correlation function (in seismic ambient noise applications) represented on the time-frequency domain in function of their coherence. I agree on that all cross-correlation are weighted equally. However, in conventional processing flows, anomalous cross-correlation are discarded in advance or in a second iteration. For example, Boué et al. (2015) classify day-long cross-correlation in high- and low-coherence days according to its coherence with the stack of one year of day-long cross-correlations and then stack them linearly. In addition, 1-bit amplitude normalization followed by cross-correlation, normalized cross-correlation or phase cross-correlation (PCC) methods contributed to severely improve the quality of the resulting EGFs by weighting signals instantaneously (1-bit & PCC) or per pair (normalized cross-correlation). Consequently, (1) it is not correct saying that these methods do not analyze SNR, and (2) it is not important that "ALL non-suitable cross-correlation functions are excluded from the final stack" as long as there are a subset of cross-correlation functions that contributes to the "final stack" with enough energy to be able to detect the signals under study.

**Authors:**

We corrected several typos in the text and tried to make it more clear.

We did not mean that these methods do not analyze SNR. We mean that these methods do not use SNR itself as an objective function for excluding of cross-correlation functions from the stack. We compared our results with weight stacking and this comparison shows that excluding as much as possible non-suitable cross-correlation functions is very important in the case when 1) one cannot continue experiment for a long time; 2) spatial distribution of noise sources is not stable during the

data acquisition period.  Such factors may be not important for studies using, for example, coda waves, or experiments using ambient noise for lithosphere studies, but they are important in applied geophysics tasks. In our example, exclusion of non-suitable functions from stack allowed to increase SNR four times.

**Referee:**

• Line 19-21: In teleseismic coda-wave interferometry the source is known indeed. Conventionally, only a few hours of data are cross-correlated at once. As the source is an earthquake using the term "distribution of noise sources" is at least misleading. You may check Pham et al. (2018) for recent work on this topic.

**Authors:**

We changed "distribution of noise sources" to "source location" in this place.

**Referee:**

• Line 23: We cannot control or change the distribution of sources; at most, what we can pursue is improving the balance of sources.

**Authors:**

Under the word "control", we mean "take into account". This sentence in the text has been changed.

**Referee:**

• Line 25: It is not correct citing Li et al. (2017) here, see comment from line 16 and lines 17-19 above.

**Authors:**

The reference has been changed.

**Referee:**

• Line 24-26: Methods weighting or selecting correlations according to its coherence (e.g., Boué et al 2015 or yours), SNR (e.g., Cheng et al., 2015 and Nakata et al., 2015), or even rms (e.g., Shirzad et al., 2014) are close related. They all seek an improvement on the quality of the signals observed in stacked interstation correlations. They only differ on the strategies used to evaluate the contribution of each correlation to build up signals in the final stack and how they are weighted. In my opinion, it is key that you explain these ideas very clearly to allow the reader judge the contribution of your method in comparison to existing alternatives and to evaluate its potential contribution in their applications.

**Authors:**

We compared SNR of EGFs after using some of these methods and our method. The result of this comparison is presented in Figure 6. Moreover, in our method the global optimization is used for obtaining the best solution for SNR. This is the key difference of our method from other methods. We described this in Introduction and in the section with the description of the method.

**Referee:**

Pg. 3, eq. (1): Your SNR estimator evaluates the signal level assuming that the signal arrives from $-t_{ds}$ to $t_{ds}$ and considers everything as noise outside this range in order to evaluate the noise level. Two things have to be discussed to justify that this estimator is sufficient for your application: what happens when strong signals (not noise) are present from $t_{ds}$ to $t_{m}$ or from $-t_{m}$ to $-t_{ds}$, and when signals are too weak to be observed in a cross-correlation function but arise after stacking.

**Authors:**

For calculation of noise level and SNR estimation, we use intervals from $-t_{ds}$ to $-t_{m}$ or from $t_{m}$ to $t_{ds}$. Of course, inside these intervals some signal also may exists. In this case, this signal is considered as an artefact and this function is excluded from stack because of low SNR. To the moment, we did not notice that rejection of such cross-correlation functions (in other words, removing of outliers) makes impact on the final solution of optimization problem. If such problem arises in the future, further improvement of noise level estimation method (for example, by using mode instead of the average) can solve the problem. To the moment, this option is not implemented because it increases significantly the calculation time. This problem is partially addressed in Discussion part of the manuscript.

We also would like to pay attention that our task was to develop a practical technique for investigating mainly relatively shallow targets (e.g. the depth range that is relevant for environmental and explorational geophysics). For such areas the range of apparent velocities in this depth range (which is one of the parameters in our algorithm) is known from a-priori information (DSS, drilling etc.). In particular, both test areas in our paper are located in the Fennoscandian Shield (e.g. not completely unknown region) so we actually can predict quite well what phases we need to look for and in what time intervals they can appear.

**Referee:**

Pg. 9 lines 18-20: Standard array methods such as beamforming allow us to measure backazimuth and apparent slowness of signals generated by punctual noise sources and then estimate the direction of arrival of surface waves or the location of body waves (e.g., through backprojection). To first order, the aperture of the array determines the resolution of these measurements. Can you be more specific on how you can relate directly the aperture of the arrays with the location of noise sources.

**Authors:**

This comment is confusing, because we were not relating the aperture of the arrays with the location of noise sources in the text. In our paper we used the classical array techniques (Rost et al., 2002; Schweizer et al., 2012) that are based on the plane-wave approximation, and array aperture is a constraint that is used for testing whether the plane-wave approximation is valid for selected wave frequency.

**Referee:**

Pg. 10 lines 3-8: The figure of using from about 8 to 35% of the number of correlations available in the final stack is important for the reader, could you add this information to the manuscript.

**Authors:**

The is added.

**Referee:**

Finally, the English usage has to further improved. For example, in pg. 3 lines 23 and 24 and in pg. 4 lines 3 and 6, using "Let it is" or "Let they are" is not correct. You may use "It is / They are" or "Let it be".

**Authors**

The language was checked**.**

REFERENCES

Phạm, T.-S., Tkalčić, H., Sambridge, M., & Kennett, B.L.N. (2018). Earth's correlation wavefield: Late coda correlation. Geophysical Research Letters, 45, 3035–3042. doi:10.1002/2018GL077244